# Video Frame Interpolation without Temporal Priors

**Youjian Zhang**\*
The University of Sydney, Australia
`yzha0535@uni.sydney.edu.au`

**Chaoyue Wang**\*
The University of Sydney, Australia
`chaoyue.wang@sydney.edu.au`

**Dacheng Tao**
The University of Sydney, Australia
`dacheng.tao@sydney.edu.au`

## Abstract

Video frame interpolation, which aims to synthesize non-exist intermediate frames in a video sequence, is an important research topic in computer vision. Existing video frame interpolation methods have achieved remarkable results under specific assumptions, such as instant or known exposure time. However, in complicated real-world situations, the temporal priors of videos, *i.e.,* frames per second (FPS) and frame exposure time, may vary from different camera sensors. When test videos are taken under different exposure settings from training ones, the interpolated frames will suffer significant misalignment problems. In this work, we solve the video frame interpolation problem in a general situation, where input frames can be acquired under uncertain exposure (and interval) time. Unlike previous methods that can only be applied to a specific temporal prior, we derive a general curvilinear motion trajectory formula from four consecutive sharp frames or two consecutive blurry frames without temporal priors. Moreover, utilizing constraints within adjacent motion trajectories, we devise a novel optical flow refinement strategy for better interpolation results. Finally, experiments demonstrate that one well-trained model is enough for synthesizing high-quality slow-motion videos under complicated real-world situations. Codes are available on `https://github.com/yjzhang96/UTI-VFI`.

## 1  Introduction

Video frame interpolation aims to synthesize non-exist intermediate frames and thereby provides a visually fluid video sequence. It has broad application prospects, such as slow motion production [13], up-converting frame rate [3] and novel-view rendering [6].

Many state-of-the-art video interpolation methods [1, 12, 17, 34] aim to estimate the object motion and occlusion with the assistance of optical flow. Through refining forward and backward motion flows among several frames, these methods can directly warp pixels to synthesize desired intermediate frames. To achieve this goal, some popular datasets, of which either triplet images or 240fps high-frame-rate videos, are collected as the ground-truth of real-world motions. Meanwhile, to evaluate the performance of proposed methods, the well-trained model is tested using frames collected in a similar way. Although significant improvement has demonstrated by experiments of recent works, people may ask if the same (or similar) performance can be achieved in complicated real-world situations.

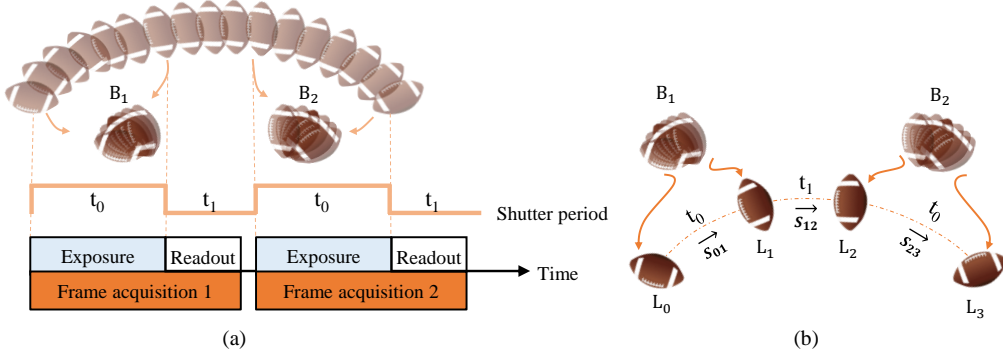

Figure 1: (a) Illustration of frame acquisition in video shooting. In real-world situation, the time interval $t_0$ and $t_1$ are unknown and may vary under different exposure setting. In the specific example in the figure, the time intervals are set to $t_0/t_1 = 6 : 4$, which indicates the intra-frame and inter-frame interpolation should be 7 and 3 frames respectively when we want to interpolate 10 frames. (b) Our proposed method aims to determine the uncertain time intervals and perform interpolation from the four consecutive states.

To comprehensively discuss this question, we first revisit the principle of video frame acquisition. As illustrated in Fig. 1 (a), the frame acquisition process usually includes two phases: exposure phase and readout phase. In the exposure phase, the shutter opens for a duration of $t_0$ so that the photosensitive sensor is exposed. In the readout phase, the camera reads the charge on the pixel array and convert the signal to get the pixel value. Considering different technologies of cameras, the readout phase could be either overlapped or non-overlapped with the exposure phase. Here, Fig. 1 (a) is an example of non-overlapped exposure. For easy discussing, we define the time interval between two exposures as $t_1$. Thus, a complete shutter period is defined as the time period $t_0 + t_1$. Correspondingly, frames per second (FPS) is defined as the reciprocal of the shutter period. Note that, $t_1$ cannot be eliminated because of the intrinsic demand of the sensor. Meanwhile, $t_0$ cannot be too short compared to shutter period, otherwise it will produce a visually discontinuous video.

The exposure time $t_0$ and the interval time $t_1$ (or FPS $\frac{1}{t_0+t_1}$) are two important parameters of a camera sensor, and they could vary largely across different cameras [4]. Therefore, when we perform frame interpolation on real-world videos, following challenges should be further considered: 1) Due to the existence of exposure time, the movement of the camera and object may produce motion/dynamic blur within a video frame. Directly performing the interpolation between blurry frames would lead to inferior visual results. A more severe blur would usually occur in the lower frame-rate video since the exposure time is relatively long. 2) Simply combining deblurring and video interpolation techniques may not handle the blurry video frames well. For blurry video frames, we should not only focus on the inter-frame interpolation, but also perform the intra-frame interpolation. 3) Note that $t_0$ and $t_1$ may vary due to the limitation of equipment or different exposure settings, the number of interpolated frames and corresponding motion trajectories will vary accordingly. For example, in the instance of Fig 1 (a), if we want to up-convert the FPS by 10 times, we should interpolate 7 frames underlying each blurry frame, and 3 frames between the two consecutive frames. Similarly, the estimation of the motion trajectory must consider the uneven time intervals. According to our observation, most existing works cannot overcome these three challenges simultaneously. Although the most recent works [13, 27] manage to solve the problem of motion blur in video interpolation, they are trained on the specific exposure setting and could be hard to generalize to different situations.

To address these issues, in this work, we consider the video frame interpolation problem in a more general situation and aim to deliver more accurate interpolation results. Specifically, giving a video sequence as the input, we first train a second-order residual key-states restoration network to synthesize the start and the end states for each frame, *e.g.* $L_0$ and $L_1$ in Fig. 1 (b). If there exists zero movement (misalignment) between two states, the video frame is regarded as one instant frame (*i.e.* without blur). Otherwise, the exposure time cannot be ignored, and both inter- and intra-interpolation are performed. Moreover, following the same assumption as [34], *i.e.* the acceleration of motion remains consistent during consecutive frames, we apply the quadratic model [18, 34] to the general video acquisition situation. We derive the general curvilinear motion representation without temporal priors from consecutive four key-states, such as $L_0$, $L_1$, $L_2$ and $L_3$ in Fig. 1 (b). Meanwhile, the

relationship between $t_0$ and $t_1$ can be determined by the displacements between key-states, *i.e.* $\boldsymbol{S_{01}}$, $\boldsymbol{S_{12}}$ and $\boldsymbol{S_{23}}$. In addition, to reduce the adverse effects caused by inferior optical-flow estimation, we further refine the optical flows with the derived trajectory priors. Finally, the refined optical flows are utilized to perform high-quality intermediate frame synthesis.

Overall, in this paper, we make following contributions: 1) We propose a restoration network to synthesize start and end states of the input video frames. This network is able to handle different exposure settings, and remove blur in the original video clip; 2) We derive a curvilinear motion representation which is sensitive to different exposure settings, thereby providing a more accurate frame alignment for uncertain time interval interpolation; 3) We further refine the optical flow with the trajectory priors to improve the interpolation results. We construct different datasets to simulate the different exposure settings in real scenarios. Comprehensive experiments on these datasets and real-world vidoes demonstrate the effectiveness of our proposed framework.

## 2   Related Works

**Video frame interpolation.** Most popular video interpolation methods utilize optical flow [12, 34, 17, 1, 2, 35] to predict the motion for the interpolated frame. Some methods [23, 22, 9] estimate space-varying and separable convolution filters for each pixel, and synthesize the interpolated pixel from a convolution between two adjacent patches. Xu *et al.* [34] proposes a quadratic interpolation to allow the interpolated motion to be curvilinear instead of being uniform and linear. However, all these methods will encounter difficulties when processing the blurry video since the optical flow/motion estimation will be inaccurate.

**Video/Image deblurring.** Conventional video deblurring methods [5, 10, 33] usually apply the deconvolution algorithm with the assistance of image priors or regulations. To make full use of adjacent frames, Hyun *et al.* [11] utilize inter-frame optical flow to estimate blur kernels. Ren *et al.* [25] also apply optical flow to facilitate the segmentation result. More recently, deep convolutional neural networks (CNN) have been applied to bypass the restriction of blur type or image priors [19, 30, 7, 20, 16, 36], and enable an end-to-end training scheme by introducing the synthetic real-world scene datasets [19, 30]. To exploit the temporal relationship, Nah *et al.* [20] propose a recurrent neural network (RNN) to iteratively update the hidden state for output frames. Wang *et al.* [32] devise a pyramid, cascading and deformable alignment module to conduct a better frame alignment in feature level, and their method won the first place in the NITRE19 video deblurring challenge [21]. There are also some works [37, 14, 24] learning to extract a video clip from a blurry image, which can be considered as a combination of image deblurring with intra-frame interpolation.

**Joint video deblurring and interpolation.** Recent methods [13, 27] have been proposed to address the blurry video interpolation problem. Among them, Jin *et al.* [13] first extract several keyframes, and then interpolate the middle frame from two adjacent frames. Meantime, Shen *et al.* [27] proposed a joint interpolation method, where they simultaneously output the deblurred frame and interpolated frame in a pyramid framework. Both these methods have pre-defined a specific setting for the blurry video exposure mechanism, which may fail when applied to a video acquired from other equipment or other camera settings.

## 3   The Proposed Video Interpolation Scheme

To address the aforementioned challenges of video frame interpolation without temporal priors, in this section, we introduce the proposed interpolation scheme in detail. Firstly, to overcome the problem caused by the uncertainty of the time interval, we derive a new quadratic formula for different exposure settings. Then, utilizing the motion flow priors contained in the formula, we further refine the estimated optical flow for more accurate time interval and trajectory estimation. Finally, we introduce the second-order residual learning strategy for key-states restoration from input frame sequences.

### 3.1   From equal time interval to uncertain time interval

To interpolate intermediate frame $L_t$ between two consecutive frames $L_1$ and $L_2$, the optical flow based video interpolation methods [12, 17, 34] aim to estimate the optical flow from frame $L_1$ to $L_t$,

or frame $L_2$ to $L_t$. Recently, inspired by [18], Xu *et al.* [34] have relaxed the constrains of motion from linear displacement to quadratic curvilinear, which corresponds to acceleration-aware motions:

$$S_{1t} = (S_{12} - S_{01})/2 \times t^2 + (S_{12} + S_{01})/2 \times t, \tag{1}$$

where $S_{ab}$ means the displacement of pixels from frame $a$ to frame $b$, and it is calculated by optical flow. In order to keep the pixel coordinates aligned in each optical flow map, the start point of these optical flows should be the same. In general, the displacements are calculated as $\hat{S}_{12} = f_{1\to2}, \hat{S}_{01} = -f_{1\to0}$, where $f_{a\to b}$ denotes the optical flow from frame a to frame b.

However, Eq.(1) is based on the equal time interval assumption. This assumption is not applicable to the general situation where the time intervals $t_0$ and $t_1$ may vary accordingly. Here, we define a shutter period as one unit time, and the ratio between $t_1$ and $t_0$ is $\lambda$, *i.e.* $t_0 + t_1 = 1$, $t_1/t_0 = \lambda$. Different from [34] which employs three neighboring frames to calculate the quadratic trajectory, we take four consecutive key-states into consideration as shown in Fig. 1 (b). Naturally, if the time intervals become unknown, four key-states (*i.e.* three flows) are requested to determine a unique quadratic movement. If we assume the acceleration remains constant from frame $L_0$ to $L_3$, then we can express $S_{01}, S_{12}$ and $S_{23}$ with velocity and acceleration:

$$2S_{12} = (2v_1 + at_1) \times t_1,$$
$$S_{01} + S_{23} = (2v_1 + at_1) \times t_0. \tag{2}$$

This equation set indicates that vector $S_{12}$ has a same direction with vector resultant $S_{01} + S_{23}$. In addition, we can derive the time interval ratio $\lambda$ as:

$$\lambda = \frac{t_1}{t_0} = \frac{2S_{12}}{S_{01} + S_{23}}. \tag{3}$$

By far, we are able to solve the $t_0$ and $t_1$ under the condition that $t_0 + t_1 = 1$. Further deriving the velocity and acceleration of the movement, we can get the expression of trajectory between frame $L_1$ and frame $L_2$:

$$S_{1t} = (\lambda + 1)(S_{23} - S_{01})/2 \times t^2 + (\lambda S_{01} + (S_{01} + S_{23})/2) \times t, t \in (t_{L_0}, t_{L_1}). \tag{4}$$

Note that when the time intervals are equal, *i.e.* $\lambda = 1$, our Eq.(4) can be degraded to Eq.(1), *i.e.* the QVI interpolation [34] is a special case of our framework.

### 3.2 Optical flow refinement

As shown in Eq.(3) (4), we can obtain flow $S_{1t}$ using the pixel displacements among four key-states. In order to keep the position aligned, $S_{23}$ should be represented as:

$$\hat{S}_{23} = f_{1\to3} - f_{1\to2}, \tag{5}$$

which denotes the movements of pixels in frame $L_1$ from time $t_{L_2}$ to $t_{L_3}$. In practical, we estimate all the optical flows using the state-of-the-art PWC-Net [31]. However, directly using flow calculated by Eq.(5) does not work well in our situation, since there may exist serious errors in two aspects: 1) the flow estimation error owing to the long time interval between frame 1 and frame 3, *i.e.* $f_{1\to3}$; 2) the pixel misalignment when we conduct vector subtraction.

Therefore, we propose a flow refinement network $\mathcal{F}_\mathcal{R}$ to acquire the refined flow $\hat{S}'_{23}$. Since it is hard to obtain the ground-truth of the target flow $S_{23}$, we employ the trajectory prior implied in Eq.(2) (3) as our penalization. Specifically, $\hat{S}_{01} + \hat{S}_{23}$ and $\hat{S}_{12}$ should have following two implicit constraints: 1) these two vectors have the same direction; 2) since $\lambda$ is a constant, the ratio of the two vectors should be uniform across the image. With these two priors as constraints, we are able to correct the value of one optical flow when the other two are fixed. Note that, although $f_{1\to0}$ and $f_{1\to2}$ are also the estimation of pixel displacements, they could deliver more accurate motion estimation than $\hat{S}_{23}$. Therefore, the refinement process can be formulated as:

$$\hat{S}'_{23} = \mathcal{F}_\mathcal{R}(f_{1\to0}, f_{1\to2}, \hat{S}_{23}). \tag{6}$$

We use a U-Net [26] with skip connections to learn the mapping from the original flow to the refined outputs. Aforementioned priors are implicitly encoded into our loss function, where we utilize $f_{0\to1}$ and $f_{1\to2}$ to constrain the output flow. The loss function is calculated as:

$$\mathcal{L}_r = |\hat{S}_{23} - (2/\lambda f_{1\to2} + f_{1\to0})|_1. \tag{7}$$

Finally, the refined $\hat{S}'_{23}$ can be substituted into Eq.(4) to compute a more accurate $S_{1t}$.

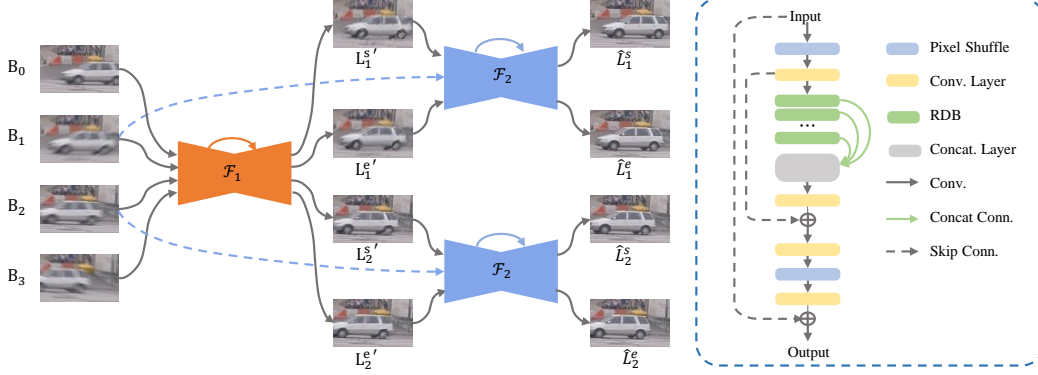

Figure 2: **Overview of our key-states restoration network.** The left figure shows the inputs and outputs of each sub-network. The right figure shows the backbone structure of $\mathcal{F}_1$ and $\mathcal{F}_2$.

### 3.3 Second-order residual learning for key-states restoration

Our principle for choosing key-states is to ensure that they are unambiguous under different exposure settings. For each input frame, we attempt to restore its instant states of the start and end of the exposure, since their physical meaning is consistent in different exposure settings. For sharp images (*i.e.* without motion blur), the start and end states should be the same. For blurry frames, the start and end states define the boundary of the motion blur, which makes them easier to restore. In addition, ours could short the temporal range for subsequent interpolation, which leads to more accurate interpolation results. More discussion can be found in the experiment section.

As shown in Fig. 2, we propose the second-order residual learning pipeline to extract the key-states from input frames. Firstly, in order to avoid the temporal ambiguity of the start and end states, four consecutive frames are fed into the network $\mathcal{F}_1$. Utilizing the implicit motion direction existed in the input sequence, the network is trained to synthesize residuals to be summed up with input blurry frames, and deliver the start and end states of $B_1$ and $B_2$. This process can be formulated as:

$$(\hat{L}_i^s, \hat{L}_i^e) = \mathcal{F}_1(B_{seq}) + B_i, \; i = 1, 2, \tag{8}$$

where $(\hat{L}_i^s, \hat{L}_i^e)$ denotes the the estimated instant start and end states respectively, and $B_{seq}$ denotes the input sequence $\{B_0, \cdots, B_3\}$.

In the experiments, although the network $\mathcal{F}_1$ achieves reasonable performance, we find it still suffers from some limitations. Firstly, the network gets four inputs to eliminate the temporal ambiguity, which decreases its deblurring capability more or less. Secondly, the fitting ability of residual is relatively poor when modeling a more severe blur. To address these issues, we further improve the deblurring performance by introducing the second-order residual learning. Specifically, we refer the Eq.(8) as first order residual, and derive the second-order residual learning as:

$$(\hat{L}_i^s, \hat{L}_i^e) = \mathcal{F}_2(B_i, \mathcal{F}_1(B_{seq}) + B_i) + \mathcal{F}_1(B_{seq}) + B_i, \; i = 1, 2. \tag{9}$$

Here, the network $\mathcal{F}_2$ aims to synthesize higher order residual of the target mapping. Since the temporal order of $\hat{L}_i^s$ and $\hat{L}_i^e$ has been initially determined by function $\mathcal{F}_1$, $\mathcal{F}_2$ can focus on restoring a pair of key-states. In experiments, this structure could improve the PSNR by around 1.5 dB.

## 4 Experiments

In this section, we introduce the datasets we used for training and test, and the training configuration of our models. Then we compare the proposed framework with state-of-the-art methods both quantitative and qualitative. Finally, we carry out an ablation study of our proposed components.

### 4.1 Datasets

To simulate the real-world situations and build datasets for more general video interpolation, we synthesize low-frame-rate videos from the sharp high-frame-rate video sequence. Considering the

video acquisition principle we discussed before, we average several consecutive frames taken by a 240fps camera to simulate one frame taken by a low-frame-rate camera. Similar to all the existing blurry image/video datasets, such synthesis is feasible if the relative motion between camera and object is not too large to produce the 'ghosting' artifacts. Meanwhile, we discard several consecutive frames to simulate the shutter closed time interval. In this way, we create videos filmed in different exposure settings by altering the number of frames averaged and discarded. Specifically, we denote the number of exposure frames as $m$, and the number of discarded frames as $n$, thus $m + n$ frames constitute a shutter period. We set $m + n = 10$ to down-sample the original 240 fps video to 24 fps, which is a common FPS setting in our daily life. For fair comparisons, we set $m$ as odd numbers ($m = 5, 7, 9$), since most other methods request the middle frame as ground-truth.

We apply the synthetic rule on both GoPro dataset [19] and Adobe240 dataset [30], and name these synthetic datasets as "dataset-m-n". Finally we get "GoPro-5-5", "GoPro-7-3", "GoPro-9-1" and "Adobe240-5-5", "Adobe240-7-3", "Adobe240-9-1" respectively. In addition, we also provide datasets "GoPro-5-3" and "GoPro-7-1" to perform a fair comparison with [27] since it can only upsample the video by multiple of 2. Noted that the other video interpolation datasets such as UCF101 [29] and Vimeo-90k [35] are not applicable for our comparison, since they only provide sharp frame triplets.

### 4.2 Implementation details

To train the key-states restoration network, we first train the network $\mathcal{F}_1$ for 200 epochs and jointly train the network $\mathcal{F}_1$ and $\mathcal{F}_2$ for another 200 epochs. To train the optical flow refinement network, 100 epochs are enough for convergence. We use Adam [15] solver for optimization, with $\beta_1 = 0.9$, $\beta_2 = 0.999$ and $\epsilon = 10^{-8}$. The learning rate is set initially to $10^{-4}$, and linearly decayed to 0. All weights are initialized using Xavier [8], and bias is initialized to 0. In total, we have 34.4 million parameters for training. In test phase, it takes 0.23s and 0.18s to run a single forward for key-states restoration network and interpolation network respectively via a NVIDIA GeForce GTX 1080 Ti graphic card.

### 4.3 Comparison with the state-of-the-art methods

**Comparison methods**. We employ two types of interpolation solution as our comparisons. The first one is the cascade model, which concatenates a deblurring model with a video frame interpolation model. Specifically, we combine the state-of-the-art image/video deblurring methods Gao *et al.*[7] and EDVR [32] with the state-of-the-art multi-frame interpolation methods QVI [34] and Super SloMo [12]. We follow the implementation of their official released code in all the experiments.

The other is the joint model of TNTT and BIN proposed by Jin *et al.* [13] and Shen *et al.* [27], respectively. These methods jointly conduct deblurring and upsampling of frame rate. Since these two methods are devised for specific exposure setting, we make some workarounds to carry out a more fair and reasonable comparison. Since the original TNTT [13] model need to iteratively interpolate the middle frame to fill the vacant indexes, we devise a specific interpolation sequence for each exposure setting, namely TNTT*. In addition, since BIN [27] is devised to up-convert the frame rate by 2 times, which shares the similiar function with our key-states restoration module, we compare their initial results with our first stage outputs. For multi-frame interpolation results, we iteratively interpolate the outputs of BIN and obtain the "8x frame rate" results. For this comparison, we prepare the dataset "5-3" and "7-1" as two different exposure setting of 30 fps video. We re-train and test the BIN model using the mixed datasets "5-3" and "7-1". Yet, our model is only trained on the mix datasets of "5-5", "7-3" and "9-1". Here, we also test our well-trained model on the "5-3" and "7-1" settings, experiments in Table 3 shows the great generalization ability of our proposed framework.

**Blurry video interpolation.** As shown in Table 1, Table 2 and Table 3, both our deblurring and overall interpolation perform favorably against former methods. In addition, several important observations can be made from these results. Firstly, in the deblurring phase, former video deblurring methods encounter great difficulties in maintaining a promising performance in our datasets with different exposure settings. For example, the original TNTT which is trained on "GoPro-9-1" performs inferior in generalizing to other test sets. Moreover, even trained on our mixed datasets, the EDVR deteriorates significantly from dataset "5-5" to datasets "7-3" and "9-1". For the final interpolation results, we can see that cascade models are sub-optimal for the overall performance. Although

Table 1: Quantitative comparison on the GoPro datasets [19].

| Method | Deblurring | | | | | | Interpolation | | | | | |
|---|---|---|---|---|---|---|---|---|---|---|---|---|
| | GoPro-5-5 | | GoPro-7-3 | | GoPro-9-1 | | GoPro-5-5 | | GoPro-7-3 | | GoPro-9-1 | |
| | PSNR | SSIM | PSNR | SSIM | PSNR | SSIM | PSNR | SSIM | PSNR | SSIM | PSNR | SSIM |
| EDVR + SloMo | 31.97 | 0.9448 | 29.60 | 0.9399 | 28.69 | 0.9225 | 27.74 | 0.9010 | 27.09 | 0.9013 | 26.71 | 0.8906 |
| EDVR + QVI | | | | | | | 28.57 | 0.9152 | 27.42 | 0.9132 | 27.21 | 0.9007 |
| Gao [7] + SloMo | 32.58 | 0.9647 | 32.64 | 0.9674 | 31.51 | 0.9586 | 28.22 | 0.9086 | 28.31 | 0.9101 | 27.97 | 0.9050 |
| Gao [7] + QVI | | | | | | | 29.13 | 0.9255 | 29.2 | 0.9263 | 28.5 | 0.9113 |
| TNTT [13] | 26.78 | 0.8934 | 28.4 | 0.9185 | 30.15 | 0.9383 | 25.29 | 0.8335 | 27.94 | 0.9052 | 30.29 | 0.9398 |
| TNTT* | 32.49 | 0.9660 | 31.45 | 0.9580 | 30.92 | 0.9526 | 28.39 | 0.8660 | 30.92 | 0.9486 | 30.82 | 0.9479 |
| Ours | **34.00** | **0.9758** | 32.63 | **0.9674** | **31.72** | **0.9597** | **32.47** | **0.9658** | **31.95** | **0.9628** | **30.95** | **0.9536** |

Table 2: Quantitative comparison on the Adobe240 datasets [30].

| Method | Deblurring | | | | | | Interpolation | | | | | |
|---|---|---|---|---|---|---|---|---|---|---|---|---|
| | Adobe240-5-5 | | Adobe240-7-3 | | Adobe240-9-1 | | Adobe240-5-5 | | Adobe240-7-3 | | Adobe240-9-1 | |
| | PSNR | SSIM | PSNR | SSIM | PSNR | SSIM | PSNR | SSIM | PSNR | SSIM | PSNR | SSIM |
| EDVR + SloMo | 31.97 | 0.9478 | 29.96 | 0.9254 | 28.49 | 0.9051 | 28.82 | 0.9204 | 27.85 | 0.9043 | 27.02 | 0.8885 |
| EDVR + QVI | | | | | | | 29.5 | 0.9291 | 28.36 | 0.9111 | 27.42 | 0.8937 |
| Gao [7] + SloMo | 29.39 | 0.9297 | 28.98 | 0.9246 | 28.45 | 0.9182 | 27.51 | 0.9057 | 27.35 | 0.9038 | 27.15 | 0.9008 |
| Gao [7] + QVI | | | | | | | 27.95 | 0.9142 | 27.77 | 0.9118 | 27.53 | 0.9080 |
| TNTT [13] | 28.75 | 0.9277 | 30.85 | 0.9381 | 29.01 | 0.9222 | 26.76 | 0.8831 | 29.10 | 0.9207 | 28.23 | 0.9148 |
| TNTT* | 32.55 | 0.9574 | 31.76 | 0.9529 | 30.91 | 0.9438 | 28.94 | 0.8836 | 31.43 | 0.9477 | 30.48 | 0.9418 |
| Ours | **34.63** | **0.9701** | **33.06** | **0.9617** | **32.21** | **0.9562** | **31.79** | **0.9565** | **31.52** | **0.9529** | **30.66** | **0.9458** |

the deblurring module achieve a high score in PSNR, there is about 3 dB loss in the following interpolation stage. This may be mainly caused by the long temporal scope between two consecutive input frames. Similar conclusion is also obtained in work of [13, 27]. On the contrary, the joint models usually can achieve a more accurate interpolation results. However, we observe that the interpolation performance of TNTT/TNTT* deteriorates heavily in the exposure setting "5-5" (from 32.49 to 28.39 for GoPro dataset). This is mainly because of the iterative synthesis of the middle frame may lead to sub-optimal results in the inter-frame interpolation. Same conclusion can be obtained from Table 3, the BIN model performs inferior when they attempt to further interpolate the middle frame between former outputs.

To intuitively visualize the comparison, we show two typical examples in Fig. 3. The first row shows that former methods fail in generating a visually clear intermediate frame. This is either because they fail in restoring a sharp frame in deblurring phase, *e.g.* EDVR [32] and TNTT [13], or the frame becomes blurry when interpolated from adjacent frames, *e.g.* Gao [7]+QVI [34], or BIN [27]. In the second row, we use Sobel operator [28] to extract the contour of interpolated results and overlap it with the contour of ground truth. Red line represents ground-truth contour and blue one means the synthesized outputs. The pinker and clearer overlapped image means a more accurate interpolation result. As we can see, our interpolated frame shows a best overlapped result with ground-truth image.

Moreover, we shot 10 real 30 FPS videos using a telephone camera, and generate the interpolated high-frame-rate video results with our method, as well as TNTT and BIN. Since there is no objective criterion to compare the generation quality, a user study is conducted for a fair comparison. According to more than 1k response collected from Amazon Mechanical Turk, there are 78.4% of people think our results are better than TNTT's, and 87.6% of people prefer ours over BIN's results. The real world video interpolation results are provided in our supplementary video.

**Uncertain time interval interpolation (sharp frames).** To futher validate the effectiveness of our proposed uncertain time interval interpolation algorithm. We compare different interpolation strategies when calculating the essential flow $S_{1t}$. To construct the videos with different time interval ratios, we sample the original high-frame-rate GoPro dataset with a different sample interval, *e.g.* to sequentially sample one frame with intervals of 6 frames and 2 frames to achieve the dataset of $\lambda = 7/3$. We compare our uncertain time interval algorithm (Model UTI) and refined version (Model UTI-refine) with original QVI [34] model, which is derived under $\lambda = 1$. We also provide a model GT as the optical flow calculated with ground-truth $\lambda$. Table 4 shows that our UTI and

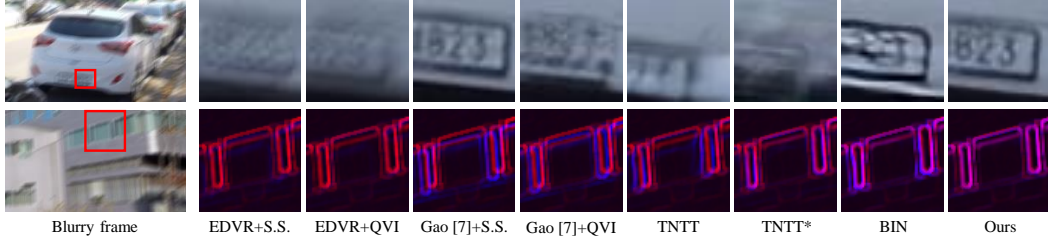

| Blurry frame | EDVR+S.S. | EDVR+QVI | Gao [7]+S.S. | Gao [7]+QVI | TNTT | TNTT* | BIN | Ours |

Figure 3: **Visual comparisons on the GoPro dataset.** Each row is an example of the interpolated results from the given blurry frames. S.S. is short for Super slomo [12]. The first row shows the original outputs. The second row is the overlap results of the interpolated frame and the ground truth. We extract the contour of each image for a better visual comparison. The red line denotes the contour of ground truth, and the blue line is the outputs of algorithms. The pinker and clearer overlapped image indicates the more accurate interpolation result.

Table 3: Quantitative comparison with BIN [27] on both GoPro and Adobe240 datasets.

|  | Method | GoPro-5-3 | | GoPro-7-1 | | Adobe240-5-3 | | Adobe240-7-1 | |
|---|---|---|---|---|---|---|---|---|---|
|  |  | PSNR | SSIM | PSNR | SSIM | PSNR | SSIM | PSNR | SSIM |
| 2x frame rate | BIN | 33.4 | 0.9649 | 32.91 | 0.9675 | 31.35 | 0.9427 | 29.91 | 0.9274 |
|  | Ours | **33.98** | **0.9771** | **32.96** | **0.9707** | **33.18** | **0.9636** | **32.65** | **0.9606** |
| 8x frame rate | BIN | 30.81 | 0.9553 | 29.14 | 0.9358 | 30.48 | 0.9402 | 29.33 | 9.9244 |
|  | Ours | **33.21** | **0.9733** | **32.3** | **0.9667** | **32.33** | **0.9611** | **31.85** | **0.9569** |

UTI-refine performs favorably to QVI model except the situation when $\lambda = 5/5$, which is owning to the optical flow estimation error in $S_{23}$. However, we can see the performance of QVI deteriorates more severe than ours when the value of $\lambda$ deviates from 1. Also, the results show that our refine network significantly improves the performance.

## 4.4 Ablation study

To see the effectiveness of our designed modules, we perform the following extensive experiments.

For the key-state restoration phase, we compare the model using different structure/input frames with the proposed model.

As we can see in Table 5, compared to the first-order residual, the model with second-order residual can increase the PSNR by around 1.5 dB. Also, the model simply cascades another stage-I's architecture, *i.e.* without $B_1$, $B_2$ as input, performs inferior to our proposed structure, suggesting the original blurry information is essential for the second-order residual learning. Both the ablation experiments show that our second-order residual is effective in refining the output of the first stage.

For the interpolation phase, we already analyzed the contribution of uncertain time interval interpolation in Table 4. Here, we evaluate the contribution of the flow refinement module. We fix the key-state restoration network and compare the interpolation outputs of the model with refinement

Table 4: Comparison of different interpolation strategies for uncertain time interval videos.

| Model | $\lambda = 5/5$ | | $\lambda = 7/3$ | | $\lambda = 9/1$ | |
|---|---|---|---|---|---|---|
|  | PSNR | SSIM | PSNR | SSIM | PSNR | SSIM |
| QVI ($\lambda = 1$) | 34.68 | 0.9820 | 32.35 | 0.9647 | 29.17 | 0.9239 |
| UTI | 33.22 | 0.9691 | 32.65 | 0.9646 | 29.75 | 0.9426 |
| UTI-refine | 34.37 | 0.9801 | 33.56 | 0.9739 | 30.57 | 0.9513 |
| GT | 34.68 | 0.9820 | 34.15 | 0.9787 | 31.34 | 0.9621 |

Table 5: Ablation study for key-state restoration and flow refinement.

| Model | | GoPro-5-5 | | GoPro-7-3 | | GoPro-9-1 | |
| | | PSNR | SSIM | PSNR | SSIM | PSNR | SSIM |
|---|---|---|---|---|---|---|---|
| Deblurring | FR | 32.39 | 0.9653 | 31.15 | 0.9547 | 30.38 | 0.9460 |
| | cascade stage-I | 32.49 | 0.9644 | 31.41 | 0.9555 | 30.76 | 0.9484 |
| | Input 2 frames | 33.07 | 0.9700 | 31.83 | 0.9620 | 30.74 | 0.9514 |
| | Proposed | **34.00** | **0.9758** | **32.63** | **0.9674** | **31.72** | **0.9597** |
| Interpolation | w/o Refine | 31.82 | 0.9586 | 31.34 | 0.9554 | 30.3 | 0.9434 |
| | Refine | **32.47** | **0.9658** | **31.95** | **0.9628** | **30.95** | **0.9536** |

(Model refine) and the model without refinement (Model w/o refine). As shown in Table 5, the model with refinement outperforms the model without refinement by around 0.6 dB. This improvement indicates that the $\hat{S}_{23}$ becomes more accurate after refinement.

# 5 Conclusion

In this work, we propose a method to tackle the video frame interpolation problem without knowing temporal priors. Taken the relationship of exposure time and shutter period into consideration, we derive a general quadratic interpolation strategy without temporal prior. We also devise a key-states restoration network to extract the temporal unambiguous sharp content from blurry frames. Our proposed method is practical to synthesize a high-frame-rate sharp video from low-frame-rate blurry videos with different exposure settings. However, there is still limitation in our work, *e.g.,* our uncertain time interval motion trajectory can only be derived when the acceleration remain constant. Though this assumption can approximate most situations in a short exposure time interval (around 1/20 s), the more challenging movement like variable acceleration motion is existing in the real scenario. We hope to relax this assumption and to have a more accurate trajectory estimation in our future works.

## Broader Impact

Video frame interpolation (VFI), which aims to overcome the temporal limitation of camera sensors, is a popular and important technology in a wide range of video processing tasks. For example, it could produce slow-motion videos without professional high-speed cameras, and it could perform the frame rate up-conversion (or video restoration) for archival footage. However, existing VFI researches can mainly apply to videos with pre-defined temporal priors, such as sharp video frames or blurry videos with known exposure settings. It may largely limit their performance in complicated real-world situations. To our best knowledge, the video frame interpolation framework we introduced in this paper made the first attempt to overcome these limitations.

Our proposed technique may potentially benefit a series of real-world applications and users. On the one hand, it could be more practical and convenient for users who want to convert their own videos to slow-motion, since they are not required to figure out the video sources, *i.e.* the complicated parameters of camera sensors. On the other hand, it could reduce the workload of VFI-related applications, *i.e.* it would not need to retrain new models for different exposure settings.

Since the video frame interpolation aims at video restoration and up-conversion (*i.e.* the output video shares the same content as the given video), our method may not cause negative ethical impact, if we do not discuss the content of the input video.

## Acknowledgments and Disclosure of Funding

This work was supported in part by the Australian Research Council Projects: FL-170100117, DP-180103424, IH-180100002 and IC-190100031.

## Footnotes

\* indicates equal contribution.

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
