[Reviews · NeurIPS 2020]

Review 1

Summary and Contributions: The paper looks into the mechanism in frame acquisition that leads to the frame blurriness and low frame rate, and propose a new perspective on the joint frame deblurring and frame interpolation for various video exposure scenarios. The state transition model is interesting and novel to resolve the joint problem. In detail, the curvilinear representation and optical flow refinement are proposed to achieve better frame qualities than the state-of-the-art method. The idea of the state transition model is somewhat novel to the community. Although the paper has provided experiment results on uncertain time interval interpolation, it is still unknown how the model will behave on real-world blurry and low-frame-rate videos. Overall, I highly recommend the paper to be accepted. Below we summarize the main advantages and flaws of this paper.

Strengths: + The paper shows the frame acquisition process when capturing moving objects, depicting the origins for motion blur and low frame rate. + The key instant frame state is decomposed from the frame capturing process and will help the lateral construction of neural networks. + The experiments are sufficient and the ablation studies have validated the effectiveness of the proposed modules and applicability to various exposure scenes.

Weaknesses: - The synthesis of the dataset may not be consistent with the actual blurry video acquisition process. Nevertheless, BIN shares the same inconsistency problem. Thus, do you ever apply your trained model to the actual videos? What are the limitations? - The model size, running speed of the proposed method is not compared with existing methods such as BIN. - The term "generalized" in the title does not suit very well to the context of this paper. Because on the opposite side of "generalized", "scenario-Specific" might come to peoples' minds? However, is there a scenario-specific video frame interpolation?

Correctness: Yes. Yes.

Clarity: Yes.

Relation to Prior Work: Yes.

Reproducibility: Yes

Additional Feedback: I think the overall idea should be summarized in to a better terminology instead of the "Generalized Video Frame Interpolation". Accordingly, if you found a good one for it, the paper should be polished too.


Review 2

Summary and Contributions: This paper addresses the video frame interpolation problem. It explicitly models the exposure time and readout time to handle the scenario under uncertain exposure time. It presents a curvilinear motion trajectory formula and a novel optical flow refinement network for better results. Experiments show that the proposed method outperforms competitive methods quantitatively on synthetic datasets. Overall, there is sufficient novelty in the problem setting and the proposed solution. However, the experiments were only performed on synthetic datasets, and the synthesis procedure is not physically correct. Without seeing how the proposed method performs on real videos, it is not easy to judge its effectiveness.

Strengths: The paper has good novelty. It observes that direct interpolation between blurry frames leads to inferior results, and a naïve combination of deblurring and interpolation cannot handle blurry videos well. In addition to analyzing the exposure time and readout time explicitly, the proposed method also generalizes the quadratic motion to the curvilinear motion and designs the optical flow refinement network. From the quantitative evaluation, the proposed method outperforms methods with a simple combination of deblurring and interpolation and those performing the two tasks jointly.

Weaknesses: The procedure for synthesizing frames with different exposure times from a 240fps camera is not physically correct. For example, for obtaining a 24fps video with the 1/48-s exposure time, the procedure averages five consecutive frames to simulate the exposure time and discard the next five frames for the readout time. Since each frame of the five frames for averaging does not have the exposure time of 1/240 second (some is for reading out). The five frames do not span a continuous exposure of 1/48 second. Averaging them cannot accurately simulate the frame with 1/48-s exposure time. The experiments were only carried out on synthetic datasets. As pointed out above, the synthesis procedure is not physically correct. It is not easy to judge how the proposed method performs on real videos. The performance improvement is more significant in the 5-5 setting, but it is much less evident for the other two settings.

Correctness: Other than the problem with the synthesis procedure, other parts appear correct to me. However, I did not check every step carefully.

Clarity: It takes some effort to comprehend the proposed method. In particular, Section 2 needs improvement.

Relation to Prior Work: It is clear to me.

Reproducibility: Yes

Additional Feedback: POST REBUTTAL COMMENTS: My main concern is about the performance of real videos since the synthesis process seems flawed. The rebuttal shows a real example and points out two real video examples in the supplementary video. The proposed method does outperform other methods significantly in these examples. I raise my score. However, it would be better if there are real examples and a discussion about the synthesis process's limitations in the paper.


Review 3

Summary and Contributions: This paper presents a method for interpolating frames that may be blurry and/or have unknown exposure times. The ideas is to train networks to recover sharp beginning and end points in time for the video frames, to compute multi-frame flow assuming constant acceleration, and then use this to interpolate the frames. They show that the training and test data need not have the same exposure times and show good results in comparison to several other methods on e few datasets.

Strengths: The strength of this approach is that it doesn't make as many assumption as previous work on the exposure times of the input frames and can effectively jointly deblur and interpolate video frames.

Weaknesses: The main weakness of this paper is that in some cases the improvements over previous work are not that large, interns of both visual results (Figure 3) and PSNR and SSIM. There is a lot of work in the area of frame interpolation, I think the approach here is not groundbreaking. Another smaller concern is how reasonable the constant acceleration assumptions is in practice. Does it hold often? I imagine it may break down in real scenarios

Correctness: The technical content of the paper appears to be correct

Clarity: The paper is well written and easy to follow

Relation to Prior Work: The previous work is discussed and compared to well

Reproducibility: Yes

Additional Feedback: I didn't follow this:"However, there is still limitation in our work, e.g. the proposed trajectory priors can only be used to refine one optical flow" Can you please expand on the limitation of this work?


Review 4

Summary and Contributions: The authors proposed a video frame interpolation method which can be generalized to different exposure time ratio. The main contributions of the paper is deriving a generalized motion trajectory, and proposing an optical flow refinement network trained with the derived constrains. The experiment results show the effectiveness of the method, and the visual quality is promising.

Strengths: The authors refine the motion trajectory computed from uncertain exposure interval. The derivation looks fine and the models implementing the new interpolation formula obtain a good visual and quantitative performance. Training the restoration model and optical flow refinement model with synthetic data is shown to be easily generalized to different settings even real inputs.

Weaknesses: Some of the concerns are below: 1. The claims of proposing a 'generalized' interpolation may be too strong. What could be the real cases which cannot be resolved by the proposed methods should also be discussed. I believe exposure setting could be only one of the problem of the poor generalization ability. 2. It's unclear to me how equation(2) in line 103 is derived. And in line 109, how equ(4) can be degraded to equ(1) given using different frames? 3. For the restoration network, it seems the network is just trying to achieve multi-frame deblurring with a two-stage process. What exactly the function of the two-stage network? Could the author show some outputs from different stages? In line 154, the improvement in dB can only come from more parameters in the model, but not the intuitive idea illustrated in the paper. Please double check it or visualize it. 4. Also for the restoration network, in line 147, what is the temporal ambiguity, and why the authors utilize four frames but not just 2 frame? In line 143, does the author mean B1 and B2? 5. More results on real videos should be reported. Currently, the results are reported on synthetic data and most previous methods do not share the same assumptions as this paper. Results on real videos in the supplementary material look fine, but related contents are missing in the paper, especially user study.

Correctness: Yes, see above.

Clarity: 1. The related work section can be placed after the introduction. 2. There are a couple of typos in the paper, i.e. -line 55: almost -synthesis should be synthesize (ctrl-F for all the typos across the manuscript) -line 82, detailed. -line 112, between --> among -line 153, temporal etc.

Relation to Prior Work: Yes, see above.

Reproducibility: Yes

Additional Feedback: The main concerns of the paper is the details of the derivation and lacking results on real images. Also the writing needs improvement. Many typos exist. ------ After reading the rebuttal and other reviewers' comments, I think some of the concerns are addressed in the make-up experiments. I encourage the authors to add more derivations and explanations to the final versions. I will increase my rating.

[Author Response · NeurIPS 2020]



Figure 1: A real video example. From left to right: Input, BIN, TNTT* and ours. Adobe Reader with flash player is recommended to watch this video (click to play). Users may need to enable the 'Preferences->3D&Multimedia->Use Flash Player ...' option.

Figure 2: User study

Table 1: Runtime and model size comparison

| Method | Runtime (s) | Parameters (million) |
|---|---|---|
| TNTT | 0.33 | 10.7 |
| BIN | 2.22 | 11.4 |
| Ours | 0.73 | 34.4 |

Table 2: Ablation study on key-states restoration network

| Model | PSNR | SSIM |
|---|---|---|
| Cascade stage-I | 30.76 | 0.9484 |
| Input 2 frames | 30.74 | 0.9514 |
| Proposed | 31.72 | 0.9597 |

Stage-I          Stage-II
Figure 3: Comparison of outputs from different stage in deblurring network.

We sincerely thank all reviewers for their constructive comments. All concerns will be addressed in the final version.

**Common concern #1**: Limitations of synthetic data, the synthesis procedure is not physically correct. **(R#1-Q1;R#2-Q1)**

**A:** Comparing with real frames, synthetic data have two limitations: 1) during the shutter open, a recorded 'sharp
ground-truth' may be blurry; 2) since the shutter close, the content missing (*i.e.,* discrete accumulation) may result in
'ghosting' artifact. However, these two flaws mainly arise when recorded a large movement, more specifically, when the
relative displacement during a shutter open/close period (*e.g.,* 1/480-s) crossed two or more pixels. Our work adopted
the same synthetic data and 'moderate' motion assumption with existing deblurring and blurry VFI methods. According
to previous studies, synthetic data is now the best choice for simulating unavailable real training pairs.

**Common concern #2**: Results on real videos and related user studies should be provided. **(R#1-Q1;R#2-Q2;R#4-Q5)**

**A:** In our supplementary video (after 1m29s), two interpolated videos of real scenes are reported. In addition, we shot
ten 10-sec real blurry videos using a telephone camera. A user study (Fig.2) collected through Amazon Mechanical
Turk shows our method achieved significant improvement on real videos. For each comparison pair, a user was asked
to select a better video. More than 1k responses are collected, and all videos were sorted randomly to avoid cheating.
Since the space limitation, we report a short video clip in Fig.1, all video results will be released with our codes.

**Common concern #3**: The improvement is more significant in the 5-5 setting; is not that large in some cases. **(R#2-Q3;R#3-Q1)**

**A:** In our experiments, TNTT*, an improved variant implemented by ourselves, is the only model that achieved
comparable results in some settings. Yet, it still faces the generalization problem that our work aims to solve. With one
well-trained model, our method showed constant superiority on both synthetic data and more challenging real videos.

**Common concern #4**: The title issue, 'generalized' does not suit very well to the context of this paper. **(R#1-Q3;R#4-Q1)**

**A:** We will replace our title as 'Video Frame Interpolation without Temporal Priors' and polish the main text accordingly.

**R#1-Q2:** Comparing the model size and running speed of the proposed methods with existing works.

**A:** As shown in Table 1, we adopted the official codes, tested all methods on the same task (8x interpolation) and the
same hardware. Note that BIN focuses on 2x interpolation, in our setting, it performed the interpolation repeatedly.

**R#3-Q2:** How reasonable the constant acceleration assumption is in practice?

**A:** We adopt this assumption from QVI, a SOTA method for sharp VFI (maintext-L90). In our setting, this assumption
only needs to hold for two consecutive blurry frames (around 1/20-s for a 30fps video). According to our experiments,
the derived curves can handle most cases. For challenging cases, we hope to relax this assumption in our future work.

**R#3-Q3:** Pls expand on the limitation of this work. (The trajectory prior can be only used to refine one optical-flow.)

**A:** When we employ the trajectory prior to refine the calculated pixel displacement $\hat{S}_{23}$, it is based on an assumption
that estimated optical-flows $f_{0\rightarrow1}$ and $f_{1\rightarrow2}$ are accurate. However, they may exist errors. In future work, we hope to
introduce a new trainable module to extract more accurate displacements (or optical-flow) for our interpolation.

**R#4-Q2:** How Eq.2 is derived; and how Eq.4 can be degraded to Eq.1?

**A:** Eq.2 is derived from the equation sets: $\{s_{12} = v_1 t_1 + \frac{1}{2} a t_1^2\}$ and $\{s_{01} = v_0 t_0 + \frac{1}{2} a t_0^2; s_{23} = v_2 t_2 + \frac{1}{2} a t_2^2; v_1 = v_0 + a t_0;$
$v_2 = v_1 + a t_1; t_0 = t_2\}$. Eq.4 can be degraded to Eq.1 when we set $\lambda = 1$ and substitute $2S_{12} - S_{01}$ for $S_{23}$ according
to Eq.3. Since space limitation, detailed derivation will be provided in the final version.

**R#4-Q3:** More discussions on the restoration network are required.

**A:** In our restoration network, the first stage mainly focuses on figuring out the frame sequence with correct temporal
order. The second stage aims to refine the output of the first stage using the proposed second-order residual structure.
As shown in Fig. 3, there exists a severe artifact in stage-I's output. In addition, the newly added ablation study
(Table 2) shows that, even employing the same amount of parameters, the model simply repeats stage-I's architecture
(*i.e.,* cascade stage-I) performs inferior to our proposed restoration network.

**R#4-Q4:** What is the temporal ambiguity; why utilize 4 frames but not 2?

**A:** For a single blurry frame, temporal ambiguity means there exist two possible outputs of the start/end states. Generally,
two or more consecutive frames are required to decide the temporal order. In both TNTT and our work, 4 input frames
are employed to reduce the ambiguity and improve deblurring results. A new ablation study is provided in Table 2.

**R#4-Q5:** Writing issues.

**A:** Thanks for pointing our typos. It should be $B_1$ and $B_2$ in L-143. We will carefully proofread and polish our draft.

[Meta-Review · NeurIPS 2020]

All four reviewers unanimously agreed that this paper proposes a simple yet effective technique to handle video frame deblurring and interpolation. The initial reviews had concerns about the synthesis process and the performance on real videos. The rebuttal successfully addressed them and promised several modifications to the paper. After the rebuttal period, two reviewers increased their ratings. Given this, I concur with the reviewers recommendations.